# Several different sequences are implicated in bloodstream-form-specific gene expression in *Trypanosoma brucei*

**Tania Bishola Tshitenge, Lena Reichert, Bin Liu¤, Christine Clayton◉ ***

Heidelberg University Center for Molecular Biology (ZMBH), Heidelberg, Germany

¤ Current address: Hebei Viroad Biotechnology Co. ltd, Shijiazhuang, China
* cclayton@zmbh.uni-heidelberg.de

## Abstract

The parasite *Trypanosoma brucei* grows as bloodstream forms in mammalian hosts, and as procyclic forms in tsetse flies. In trypanosomes, gene expression regulation depends heavily on post-transcriptional mechanisms. Both the RNA-binding protein RBP10 and glycosomal phosphoglycerate kinase PGKC are expressed only in mammalian-infective forms. RBP10 targets procyclic-specific mRNAs for destruction, while PGKC is required for bloodstream-form glycolysis. Developmental regulation of both is essential: expression of either RBP10 or PGKC in procyclic forms inhibits their proliferation. We show that the 3'-untranslated region of the *RBP10* mRNA is extraordinarily long—7.3kb—and were able to identify six different sequences, scattered across the untranslated region, which can independently cause bloodstream-form-specific expression. The 3'-untranslated region of the *PGKC* mRNA, although much shorter, still contains two different regions, of 125 and 153nt, that independently gave developmental regulation. No short consensus sequences were identified that were enriched either within these regulatory regions, or when compared with other mRNAs with similar regulation, suggesting that more than one regulatory RNA-binding protein is important for repression of mRNAs in procyclic forms. We also identified regions, including an AU repeat, that increased expression in bloodstream forms, or suppressed it in both forms. Trypanosome mRNAs that encode RNA-binding proteins often have extremely extended 3'-untranslated regions. We suggest that one function of this might be to act as a fail-safe mechanism to ensure correct regulation even if mRNA processing or expression of *trans* regulators is defective.

## Author summary

The parasite *Trypanosoma brucei* causes sleeping sickness in humans, and nagana in cattle, and is transmitted by Tsetse flies. It grows in the bloodstream and tissue fluids of mammalian hosts, as "bloodstream forms", and as "procyclic forms" in the midgut of tsetse flies. Several hundred proteins are expressed in a stage-specific fashion, and this is essential for parasite survival in the different environments. RBP10 is an RNA-binding protein

**Data Availability Statement:** The Western and Northern blots and original data are to be found at https://figshare.com/articles/figure/Full_Northern_and_Western_blots_RBP10_Fig_1_pdf/17099168

https://figshare.com/articles/figure/Northern_blots_RBP10_3_-UTR/17085989 https://figshare.com/articles/dataset/CAT_activity_and_mRNA_measurements_RBP10/19188431 https://figshare.com/articles/figure/Complete_Western_blots_for_PGKC_3_-UTR_assays/19188455 https://figshare.com/articles/dataset/qPCR_for_PGKC_3_UTR_analysis/19188248 https://figshare.com/articles/figure/Complete_Northern_blots_PGKC/19181888

**Funding:** This work was partially supported by Deutsche Forschungsgemeinschaft grant number CI112/28 to CC. The funders had no role in study design, data collection and analysis, decision to publish, or preparation of the manuscript.

**Competing interests:** The authors have declared that no competing interests exist.

that is expressed only in bloodstream forms. It binds to procyclic-specific mRNAs, and causes their destruction. PGKC is an enzyme that is also specifically expressed in bloodstream forms. Developmental regulation of both is essential: expression of either RBP10 or PGKC in procyclic forms prevents their growth. The mRNAs encoding both proteins are very unstable in procyclic forms, and the sequences responsible are in an "untranslated region" of the mRNA—sequences that follow the part that codes for protein. We here show that the mRNA encoding PGKC has two regions that independently cause developmental regulation, and that the very long untranslated region of the *RBP10* mRNA has no fewer than six regulatory regions, but there were no obvious similarities between them. We suggest that the presence of several different regulatory sequences in trypanosome mRNAs might be a fail-safe mechanism to ensure correct regulation.

## Introduction

Kinetoplastids are unicellular flagellates, some of which are parasites of vertebrates, invertebrates and plants. The African trypanosome *Trypanosoma brucei* is a kinetoplastid that causes sleeping sickness in humans in Africa and infects livestock throughout the tropics, with a substantial economic impact [1]. *T. brucei* are transmitted by a definitive host, the Tsetse fly, or during passive blood transfer by biting flies. The parasites multiply extracellularly as long slender bloodstream forms in the mammalian blood and tissue fluids, escaping the host immune response by expressing different Variant Surface Glycoproteins (VSGs) [2]. High cell density triggers growth arrest and a quorum sensing response, prompting differentiation of the long slender bloodstream forms to stumpy forms [3]. The stumpy form is pre-adapted for differentiation into the procyclic form, which multiplies in the Tsetse midgut. The transition from the mammalian host to the Tsetse fly entails a decrease in temperature, from 37˚C to between 20˚C and 32˚C; and a switch from glucose to amino acids as the main source of energy, with a concomitant change to dependence on mitochondrial metabolism for energy generation [4,5]. Meanwhile, the VSG coat is replaced by the procyclins [6,7]. Procyclic forms later undergo further differentiation steps before developing into epimastigotes, and then mammalian-infective metacyclic forms in the salivary glands [8]. The developmental transitions in the *T. brucei* life cycle are marked by extensive changes in mRNA and protein levels [9–17]. Differentiation of trypanosomes from the bloodstream form to the procyclic form can be achieved *in vitro* through addition of 6 mM cis-aconitate and a temperature switch from 37˚C to 27˚C, followed by a medium change [18,19].

Bloodstream-form trypanosomes rely on substrate-level phosphorylation for ATP generation, and the first seven enzymes of glycolysis and glycerol metabolism are located in a microbody, the glycosome [20,21]. Within the branch of the pathway that has pyruvate as the endproduct, the last enzyme that is in the glycosome is phosphoglycerate kinase (PGK). In procyclic-form trypanosomes, although the glycosomal glycolytic pathway is retained, it has some other roles and most phosphoglycerate kinase activity is in the cytosol. The trypanosome genome encodes three PGK isozymes [22], all of which are enzymatically active [23–25]. PGKA (Tb927.1.720) is expressed at low levels and is in the glycosome [26]; PGKB (Tb927.1.710) is the cytosolic enzyme found in procyclic forms; and PGKC (Tb927.1.700) is the glycosomal isozyme that is found in mammalian-infective forms [27,28]. This regulated compartmentation is essential. Expression of cytosolic PGK is lethal to bloodstream-form trypanosomes [29], perhaps because it disrupts the ATP/ADP balance within the glycosome

[30,31]; and expression of PGKC in procyclic forms is similarly lethal (at least in the presence of glucose), probably because it reduces cytosolic ATP production [32].

In Kinetoplastids, nearly all protein-coding genes are arranged in polycistronic transcription units. Mature mRNAs are generated from the primary transcript by 5'-*trans*-splicing of a 39nt capped leader sequence, and by 3'-polyadenylation (reviewed in [33,34]). The detection of intergenic RNA from the *PGK* locus was indeed one of the early pieces of evidence for polycistronic transcription [35]. Polyadenylation is not very precise: sites are found in a region that is about 140nt upstream of the *trans*-splice site that is used to process the following mRNA [11,36], and quite often, alternative splice acceptor sites (and therefore polyadenylation sites) are used. The parasite regulates mRNAs mainly by post-transcriptional mechanisms, supplemented, in the case of some constitutively abundant mRNAs, by the presence of multiple gene copies. Regulation of mRNA processing, degradation, and translation are therefore central to parasite homeostasis, and for changes in gene expression during differentiation [14–17]. The sequences required for regulation of mRNA stability and translation often lie in the 3'-UTRs of the mRNAs, and most regulation so far has been found to depend on RNA-binding proteins [37,38].

RBP10 (Tb927.8.2780) is an RNA-binding protein which has been detected only in bloodstream forms and metacyclic forms. The *RBP10* mRNA is also at least four times more abundant in bloodstream forms than in procyclic forms [9,10,39–41]; it persists in stumpy forms [42–44], but decreases rapidly after induction of differentiation [45]. Within the Tsetse fly, *RBP10* mRNA may be up-regulated in metacyclic forms [9,10,46] although this was not seen in all studies. Procyclic forms can be induced to differentiate to epimastigotes, and then mammalian-infective metacyclic forms, by induced expression of the RNA-binding protein RBP6 (Tb927.3.2930) [46,47]. RBP10 protein is detected in bloodstream forms [39] and RBP6-induced metacyclic forms [40,47]. Using mass spectrometry, the degree of RBP10 regulation is difficult to calculate because of poor detectability in procyclic forms, but was estimated in one study at 25-fold [48]; in another, RBP10 protein was undetectable in stumpy forms [13]. RBP10 specifically associates with procyclic-specific mRNAs that contain the motif UA (U)$_6$ in their 3'-UTRs, targeting them for destruction [49]. Depletion of RBP10 from bloodstream forms gives cells that can only grow as procyclic forms [49] and RBP10 expression in procyclic forms makes them able to grow only as bloodstream forms, with expression of some metacyclic-form *VSG* mRNAs but no detectable formation of epimastigotes [49,50]. After RBP6 induction in procyclic forms, RBP10 was also required for expression of metacyclic *VSG* mRNAs [46]. Correct developmental regulation of RBP10 is therefore critical throughout the parasite life cycle.

The distribution of 3'-UTR lengths in trypanosomes is remarkably broad. Initial estimates from high-throughput RNA sequencing (RNA-Seq) suggested median lengths of 400-500nt [11,36] and 6% over 2kb. Subsequent studies of individual genes have revealed that these are under-estimates: some trypanosome mRNAs have 3'-UTRs of 5kb or more. This is rather unusual compared with Opisthokont model organisms. For example, *Saccharomyces cerevisiae* 3'-UTRs are 0-1461nt long, with a median of 104nt [51], while *Caenorhabditis elegan*s 3'-UTRs have a median length of 133nt; less than 2% of *C. elegans* mRNAs had 3'-UTRs longer than 2kb [52]. Interestingly, the longest trypanosome 3'-UTRs tend to be found in mRNAs encoding RNA-binding proteins and protein kinases [37]. Since these protein classes have vital regulatory functions, it is likely that their expression has to be particularly tightly controlled. For example, the mRNA encoding the procyclic-specific mRNA binding protein ZC3H22 (Tb927.7.2680) has a 3'-UTR that is over 5kb long, with 9 copies of the UA (U)$_6$ motif [53].

Over 400 *T. brucei* mRNAs are at least 10-fold better expressed in bloodstream forms than in procyclic forms, as judged by the relative numbers of ribosome footprints, which measure

of both mRNA abundance and translation [14]. To our knowledge, however, the only short RNA motif that has so far been implicated in such regulation is an 8mer (UGCUACUU) that is specific to the 3'-UTR of the *VSG* mRNA [54]. In a previous study, we used a reporter assay to examine the functions of various segments of the *PGKC* 3'-UTR [55]. Results from deletions indicated that the sequences that were required for developmental regulation were in the terminal 424nt of the *PGKC* 3'-UTR [55]—a region that is likely to be bound by several different proteins [56]. In this paper, we aimed to find shorter sequences that are responsible for the bloodstream-form-specific expression of both the *PGKC* and *RBP10* mRNAs. We found that *PGKC* has at least two such sequences, while *RBP10* has at least six which are scattered throughout the 7.3 kb 3'-UTR. Our results suggest that several different sequence motifs—and therefore, probably, a similar number of RNA-binding proteins—are implicated in controlling bloodstream-form-specific mRNA stability and translation.

## Materials and methods

### Trypanosome culture

The experiments in this study were carried out using the pleomorphic cell line EATRO 1125 [57], constitutively expressing the tetracycline repressor. The bloodstream form parasites were cultured as routinely in HMI-9 medium supplemented with 10% heat inactivated foetal bovine serum at 37˚C with 5% $CO_2$. During proliferation, the cells were diluted to $1x10^5$ cells/ml and maintained in density between $0.2–1.5x10^6$ [58]. To preserve the pleomorphic morphology between the experiments, the EATRO 1125 cells were maintained in HMI-9 medium containing 1.1% methylcellulose [59]. For generation of stable cell lines, $\sim 1–2 \times 10^7$ cells were transfected by electroporation with 10 μg of linearized plasmid at 1.5 kV on an AMAXA Nucleofector. Selection of new transfectants was done after addition of appropriate antibiotic and serial dilution, usually about 6-8h after transfection. The differentiation of bloodstream forms to procyclic forms was induced by addition of 6mM *cis*-aconitate (Sigma) to $1x10^6$ long slender trypanosomes; after 17-24h, the cells were transferred into procyclic form media ($\sim 8x10^5$ cells/ml) and maintained at 27˚C without $CO_2$.

### Plasmid constructs

To assess the role of the *RBP10* 3'-UTR at full length, a cell line expressing a chloramphenicol acetyltransferase (*CAT*) mRNA with the *RBP10* 3'-UTR was generated by replacing one allele of the *RBP10* coding sequence with the coding region of the *CAT* reporter. The *RBP10* 5'-UTR was also replaced with the beta-tubulin (*TUB*) 5'-UTR (Fig 1E). To map the regulatory sequences, plasmids used for stable transfection were based on pHD2164, a dicistronic vector containing the *CAT* and neomycin phosphotransferase (*NPT*) resistance genes (Fig 2A). Downstream of the *CAT* gene, we cloned different fragments of the *RBP10* 3'-UTR in place of the actin (*ACT*) 3'-UTR, using *Sal* I and *Xho* I restriction sites (Fig 2A). The different fragments were obtained by PCR using genomic DNA from Lister 427 trypanosomes as the template. Mutations on smaller fragments of the *RBP10* 3'-UTR were done using either site directed mutagenesis (NEB, Q5 Site-Directed Mutagenesis Kit Quick Protocol, E0554) or by PCR mutagenesis with Q5 DNA polymerase.

 To study the *PGKC* 3'-UTR, the starting plasmid used, pHD3261 (Fig 3A), was built by incorporating various fragments either from previous plasmids [54], or made by PCR amplification with plasmid templates. The starting plasmid contains 790nt of the *PGKC* 3'-UTR, re-amplified from [60]. Most of the sequence, apart from the plasmid backbone, was verified by sequencing and this is included as S2 Text. Deletions were done by specific PCR followed by cloning between the *Bam* HI and *Sal* I sites (Fig 3). Some deletions occurred during the PCR,

so the 3'-UTR sequences used are included as supplementary S1 Fig. The reconstructed sequence of the derivative with the 3'-UTR of genes encoding actin (*ACT*) is S2 File.

The precise details of the different constructs and their associated primers used for cloning are included in S1 Table.

## RNA analysis

Total RNA was isolated from approximately $1x10^8$ bloodstream-form trypanosomes or $5x10^7$ procyclic-form cells growing in logarithmic phase (less than about $8 \times 10^5$/ml for bloodstream forms, or $4 \times 10^6$ /ml for procyclic forms) using either peqGold Trifast (PeqLab) or RNAzol RT following the manufacturer's instructions.

To detect the *CAT* or *CFP* mRNAs by Northern blot, 5 or 10 μg of total RNA was resolved on formaldehyde agarose gels, transferred onto nylon membranes (GE Healthcare) by capillary blotting and fixed by UV-crosslinking. The membranes were pre-hybridized in 5x SSC, 0,5% SDS with 200 mg/ml of salmon sperm DNA (200 mg/ml) and 1x Denhardt's solution, for an hour at 65˚C. The probes were generated by PCR of the coding sequences of the targeted mRNAs, followed by incorporation of radiolabelled [$\alpha^{32}$P]-dCTP and purification using the QIAGEN nucleotide removal kit according to the manufacturer's instructions. The purified probes were then added to the prehybridization solution and the membranes were hybridized with the respective probes at 65˚C for overnight (while rotating). After rinsing the membranes in 2x SSC buffer/0.5% SDS twice for 15 minutes, the probes were washed out once with 1x SSC buffer/0.5% SDS at 65˚C for 15 minutes and twice in 0.1x SSC buffer/0.5% SDS at 65˚C each for 10 minutes. The blots were then exposed onto autoradiography films for 24–48 hours and the signals were detected with the phosphorimager (Fuji, FLA-7000, GE Healthcare). Care was taken to ensure that signals were not over-exposed so that quantitation would be in the linear range. The signal intensities of the bands were measured using ImageJ. *CFP* mRNA levels were measured by quantitative real-time PCR as previously described [61].

To measure mRNA half-lives, mRNA transcription and *trans*-splicing were simultaneously inhibited by addition to the growth culture medium of 10 μg/ml Actinomycin D and 2 μg/ml Sinefungin. The cells were collected at the indicated different time points and RNA was isolated by Trizol extraction [62]. The mRNA levels were assessed by Northern blotting.

## CAT assay

To perform the CAT assays, approximately $2 \times 10^7$ cells were harvested at 2300 rpm for 8 minutes and washed three times with cold PBS. The pellet was re-suspended in 200 μl of CAT buffer (100mM Tris-HCl pH 7.8) and lysed by freeze-thawing three times using liquid nitrogen and a 37˚C heating block. The supernatants were then collected by centrifugation at 15,000×g for 5 min and kept in ice. The protein concentrations were determined by Bradford assay (BioRad) according to the manufacturer's protocol. For each setup, 0.5 μg of protein in 50 μl of CAT buffer, 10 μl of radioactive butyryl CoA ($^{14}$C), 2 μl of chloramphenicol (stock: 40 mg/ml), 200 μl of CAT buffer and 4 ml of scintillation cocktail were mixed in a Wheaton scintillation tube HDPE (neoLab #9–0149) and the incorporation of radioactive acetyl group on chloramphenicol was measured using a Beckman LS 6000IC scintillation counter.

## Western blots

Protein samples were collected from approximately $5x10^6$ cells growing at logarithmic phase. Samples were run according to standard protein separation procedures using SDS-PAGE. The primary antibodies used in this study were mouse monoclonal IgG against GFP (Santa Cruz Biotechnology) and rat α-ribosomal protein S9 (own antibody). We used horseradish

peroxidase coupled secondary antibodies (α-rat, 1:2000 and α-mouse, 1:1000). The blots were developed using an enhanced chemiluminescence kit (Amersham) according to the manufacturer's instructions. The signal intensities of the images were quantified using Image J.

## Results

### The *RBP10* 3'-UTR is sufficient for developmental regulation

Our first experiments were designed to check high-throughput results concerning RBP10 regulation. Results from RNA-Seq and ribosome profiling [17] suggest that the 3'-UTR is about 7.3 kb long, giving a total mRNA length of about 8.5 kb (Fig 1A). The middle region of the 3'-UTR is shown as being present more than once in the TREU927 genome assembly, as indicated by the grey-coloured reads in the alignment in Fig 1A, but we have been unable to locate this other copy in either this or the other available genomes (e.g. [63]). In our experiments we used the EATRO1125 *T. brucei* strain, which is differentiation-competent but for which an assembled genome [64] became available only after this work was complete.

Previous transcriptome and ribosome profiling results indicated that there are about 4 copies of *RBP10* mRNA per cell in bloodstream forms, and slightly under 1 per cell in procyclic forms [15], but that the ribosome density on the coding region is 9 [16] or 100 [14] times higher in bloodstream forms. Northern blot results for the EATRO1125 strain showed an *RBP10* mRNA that migrated slower than the 6kb marker (Fig 1B). When we extrapolated the marker curve, the results gave an *RBP10* mRNA length consistent with the 8.5 kb from RNA-Seq. In this experiment, there was 8-fold more *RBP10* mRNA in bloodstream forms than in procyclic forms (Fig 1B). The amount of RBP10 protein decreased about 3-fold after 24h incubation with 6mM *cis*-aconitate at either 20˚C or 27˚C, and both *cis*-aconitate and the temperature drop were required for the regulation (Fig 1C). Previous RNA-Seq measurements suggested an *RBP10* mRNA half-life of just over 1h in Lister 427 bloodstream forms [15]. Three new individual measurements in EATRO1125 were difficult to interpret because of an initial apparent increase in the mRNA, but did suggest a half-life of 1–2 h (Fig 1D). The initial increase has previously been seen for other stable mRNAs and its cause is unknown. For procyclic forms, the RNA-Seq replicates for degradation measurements were very poor [15], but from Northern blotting the half-life was probably less than 30 min (Fig 1D).

Finally, to find out whether the *RBP10* 3'-UTR was sufficient for regulation, we integrated a chloramphenicol acetyltransferase (*CAT*) gene into the genome of strain EATRO1125 bloodstream forms, directly replacing one *RBP10* allele (Fig 1E). After differentiation to procyclic forms, *CAT* mRNA was about 3-fold down-regulated, but there was no detectable CAT activity (Fig 1E). This shows that the *RBP10* 3'-UTR is sufficient for developmental regulation.

### The *RBP10* 3'-UTR contains numerous regulatory elements

In order to find sequences that contribute to the stability and translation of *RBP10* mRNA in bloodstream forms, or to its instability and translational repression in procyclic forms, we made use of a reporter plasmid that integrates into the tubulin locus, resulting in read-through transcription by RNA polymerase II. The *CAT* reporter mRNA has a 5'-UTR and splice signal from an *EP* procyclin gene (Fig 2A). At the 3'-end between *CAT* and *NPT* is an intergenic region from between the two actin (*ACT*) genes, with a restriction site exactly at the mapped polyadenylation site (Fig 2A). Polyadenylation of the *CAT* mRNA is directed by the polypyrimidine tract that precedes the *ACT* 5'-UTR. Cell lines can be selected with G418 using the *NPT* (neomycin phosphotransferase) marker, the mRNA of which has an *ACT* 5'-UTR. The reporter produces a *CAT* mRNA bearing the *ACT* 3'-UTR, with polyadenylation driven by the downstream splice site for *NPT*. In all experiments, the reporter plasmid was transfected into

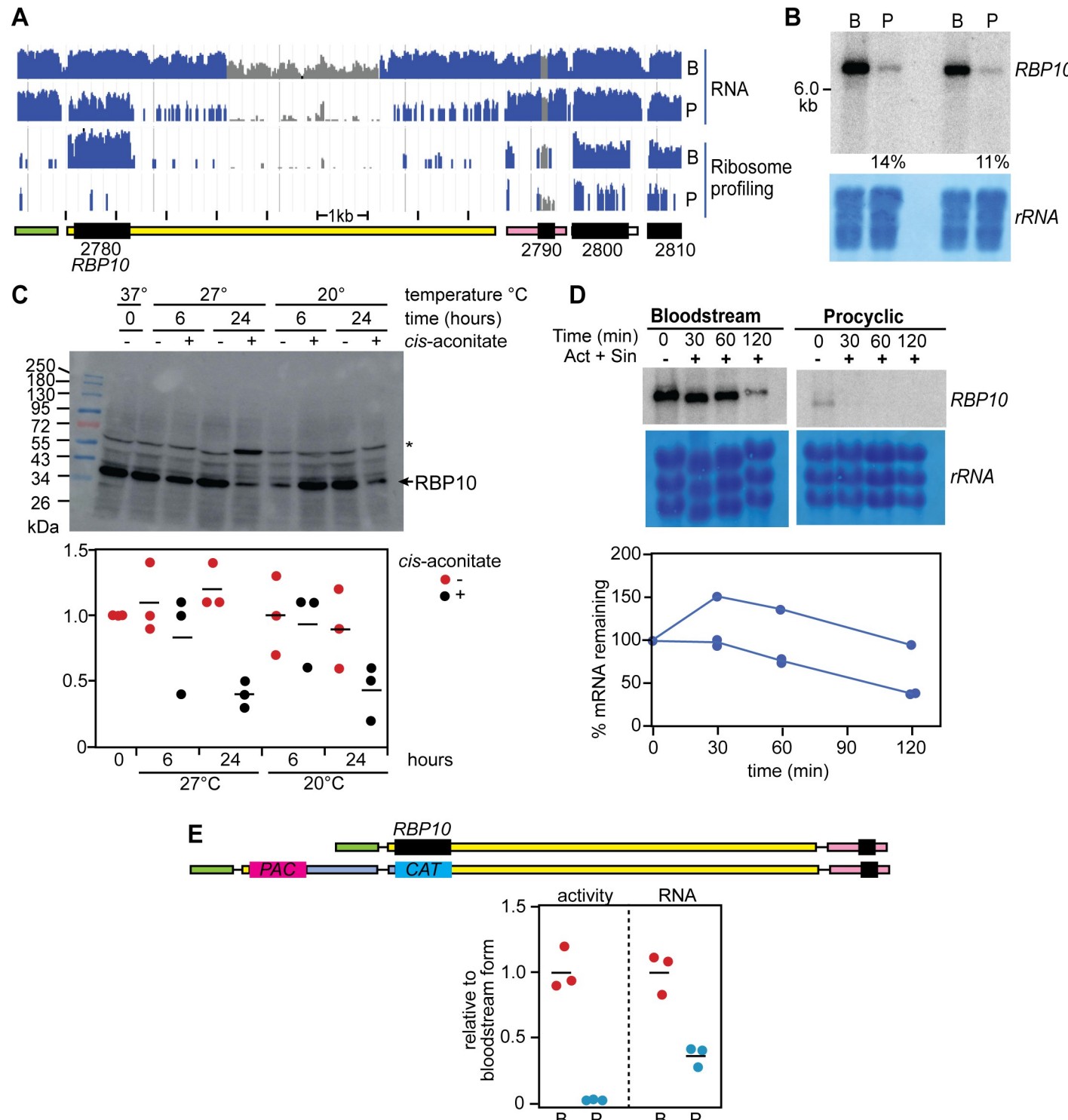

**Fig 1. Developmental regulation of the *RBP10* mRNA. A.** RNA-sequencing data and ribosome profiling [14] reads for bloodstream forms (B) and procyclic forms (P), aligned to the relevant segment of the TREU927 reference genome. Unique reads are in blue and non-unique reads are in grey, on a log scale. The sequences in the *RBP10* (Tb927.8.2780) 3'-UTR designated as non-unique in this image are also present in an TREU927 DNA contiguous sequence that has not been assigned to a specific chromosome; this segment is absent in the Lister427 (2018) genome. The positions of open reading frames (black) and untranslated regions have been re-drawn, with the initial "Tb927.8" removed for simplicity. Transcription is from left to right. Data from all remaining panels are for the EATRO1125 strain. **B.** Northern blot for two independent cultures showing the relative mRNA abundance of *RBP10* mRNA in bloodstream forms (B) and in procyclic forms (P). A section of methylene blue staining is depicted to show the loading and the measured amounts in procyclic forms, relative to bloodstream forms, are also shown. **C.** Regulation of

RBP10 protein expression during differentiation. Cells were incubated with or without 6 mM *cis*-aconitate at the temperatures and for the times indicated. The asterisk shows a band that cross-reacts with the antibody, and was used as a control. The graph shows quantitation (individual results) from three independent experiments, the horizontal line is the arithmetic mean. **D.** Half-life of *RBP10* mRNA. Cells were incubated with Actinomycin D (10 μg/ml) and Sinefungin (2 μg/ml) to stop both mRNA processing and transcription. The amount of *RBP10* mRNA in bloodstream forms was measured in three independent replicates. The initial delay or even increase in abundance seen in one replicate is commonly seen for relatively stable trypanosome mRNAs, and is of unknown origin. **E.** The *RBP10* 3'-UTR is sufficient for regulation. A dicistronic construct mediating puromycin resistance (*PAC* gene) and encoding chloramphenicol acetyltransferase (*CAT* gene) replaced one of the two *RBP10* open reading frames (upper panel) in bloodstream forms (B). These then were differentiated to procyclic forms (P). The lower panel shows measurements of CAT activities and mRNA for three independent cell lines, normalised to the average for bloodstream forms. Data for this Figure are at: https://figshare.com/articles/figure/Full_Northern_and_Western_blots_RBP10_Fig_1_pdf/17099168.

EATRO1125 bloodstream forms and two or three independent clones were then differentiated into procyclic forms. CAT activities were measured enzymatically and mRNA levels were measured by Northern blotting, which simultaneously allowed us to check the sizes of the mRNAs. All values were normalised to arithmetic mean results from the *ACT* 3'-UTR control. The sizes

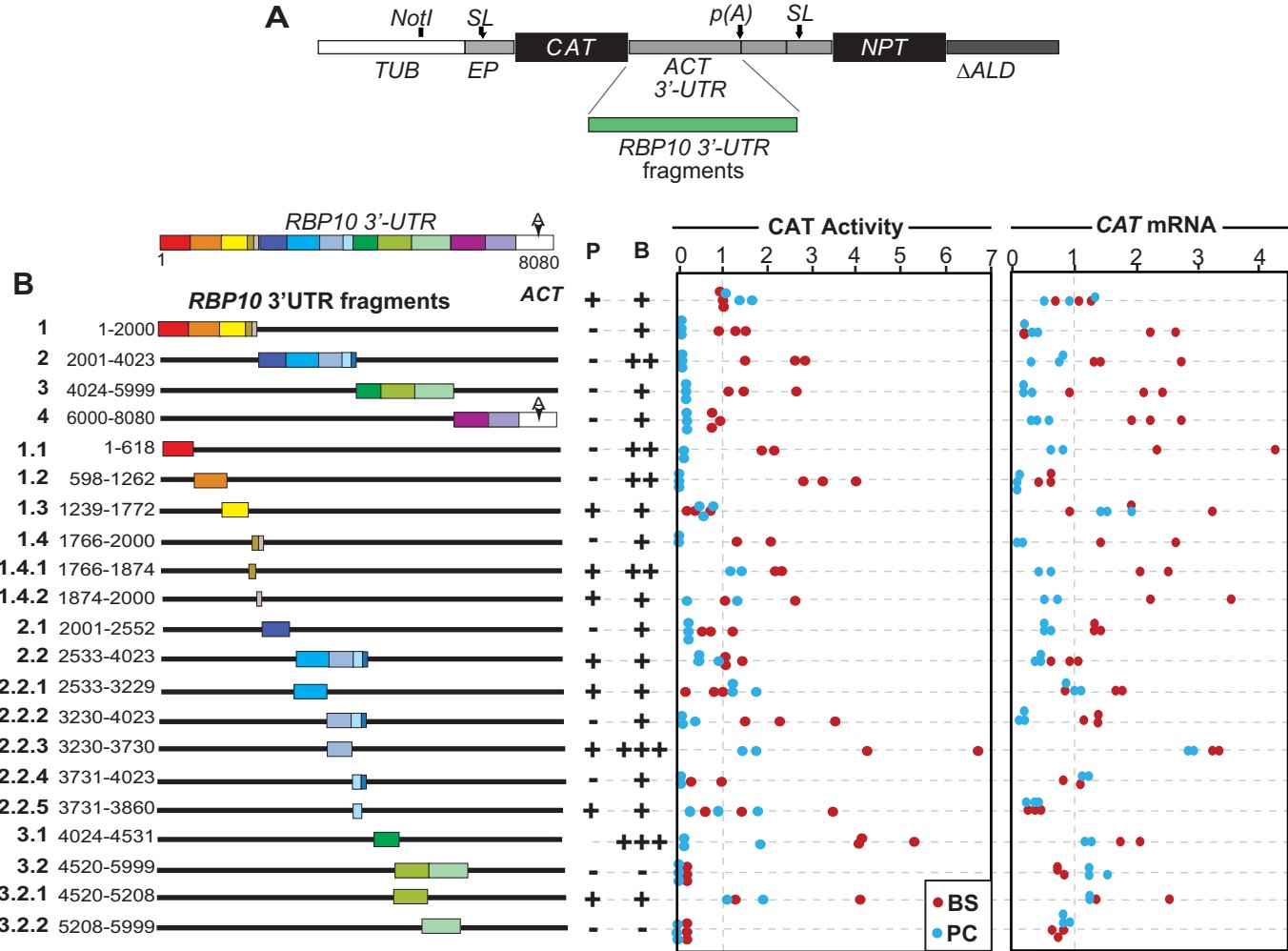

**Fig 2. The *RBP10* 3'-UTR contains several regulatory sequences. A.** Cartoon showing the *CAT* reporter construct (pHD2164) used in this study. The different fragments of the *RBP10* 3'-UTR were cloned downstream of the *CAT* reporter coding sequence. The polyadenylation site "p (A)" is specified by the *NPT* splice signal. *SL* indicates the spliced leader addition sites. *ACT* denotes actin. **B.** Schematic diagram of the *RBP10* 3'-UTR segments used to map the regulatory sequences. Different segments are in different colours and the diagram is to scale. Measurements of CAT activity and mRNA levels are on the right. Each dot is a result for an independent clone, with blue and red representing bloodstream (B) an procyclic (P) forms, respectively. Values were normalized to those for the *ACT* 3'-UTR control (pHD2164). The average result for the control BSF cell line was set to 1. For "expression", averages are: "-" = <0.25x, "+/-" = 0.25–0.5x, "+" = 0.5-2x, "++" = 2-3x and "+++" >3x. The complete Northern blots are available at: https://figshare.com/articles/figure/Northern_blots_RBP10_3_-UTR/17085989 and the measurements for this and Fig 4 are at: https://figshare.com/articles/dataset/CAT_activity_and_mRNA_measurements_RBP10/19188431.

of the mRNAs are tabulated in S2 Table and the sequences are in S1 Text. Most of the reporter mRNAs migrated either as expected, or slightly faster. The latter suggests polyadenylation upstream of the expected sites.

First, we examined four different fragments (1–4 in Fig 2B), each roughly 2 kb long. Fragment 4 extends beyond the *RBP10* polyadenylation site, including the intergenic region before downstream gene (Tb927.8.2790); and the size of the resulting RNA suggested use of the genomic processing signals (S2 Table). It was notable that the results for RNA were much more variable than for the CAT activity. Since RBP10 protein also shows more regulation than the RNA, the following discussion will consider mostly the CAT activity. We consider CAT activity to be "low" ("-" in Fig 2) if the median value was less than 25% of the activity measured using our standard, the 3'-UTR from an actin gene (*ACT*), and "high" ("++" in Fig 2) if the median CAT activity was more than twice the control; results in between are designated "+". The degree of developmental regulation is calculated by dividing the lowest value for bloodstream forms by the highest value in procyclic forms. We did not calculate P-values because with 3 replicates, it is impossible to tell whether the values are normally distributed; and we here discuss only the fragments that gave similar results for all replicates. To our surprise, all of the 2kb fragments gave low CAT activity in procyclic forms (Fig 2B). Fragment 2 also reproducibly gave high CAT activity in bloodstream forms (Fig 2B). Clearly, several regulatory sequences were present. We next further dissected the fragments. We attempted to examine sequences of equal sizes, but were constrained by the need to design PCR primers that avoided the numerous low-complexity sequences in the *RBP10* 3'-UTR. Our results showed that fragments 1.1, 1.2, 1.4 and 2.2.2 (Fig 2B) each gave low CAT activity in procyclic forms with at least 4-fold developmental regulation; fragment 1.2 also reproducibly gave more activity in bloodstream forms than the parent sequence. Fragment 1.4 (234nt) was the shortest that gave this pattern; when it was cut in half, regulation was lost, either because we accidentally cut within a relevant motif, or because the cleavage adversely affected a required secondary structure. Fragments 3.2 and 3.2.2 gave low CAT activity in both stages although the mRNA was readily detected, suggesting possible translation repression. Fragments 2.1 and 2.2.4 suppressed CAT activity in procyclics but with rather low CAT activity in bloodstream forms and little RNA regulation, suggesting a role in repression of translation in procyclic forms.

It was notable that sometimes, fragmentation of a sequence revealed activity that had been absent previously: for example, fragment 2.2 gave no regulation but the sub-fragment 2.2.2 regulated like fragment 2. Fragment 2.2 may also include a repressive element whose removal results in high activity in bloodstream forms (fragment 2.2.3). Fragment 3.1 also gave high activity in bloodstream forms. (Fig 2B). These results show that regulation of RBP10 expression is achieved by numerous sequence elements.

## At least two segments of the *PGKC* 3'-UTR contain regulatory elements

To study regulation by the *PGKC* 3'-UTR, we used the reporter plasmid shown in Fig 3A. After restriction enzyme cleavage and transfection, the plasmid integrates into the tandemly repeated alpha-beta tubulin array, cleanly replacing an alpha tubulin gene (Fig 3A). It is therefore, like the previous reporter, transcribed by RNA polymerase II. A cyan fluorescent protein (*CFP*) reporter open reading frame is followed by a 3'-UTR, then an intergenic region (IGR), splice signal and 5'-UTR from the actin (*ACT*) locus. Splicing of the mRNA encoding CFP is directed by the alpha-tubulin signal, resulting in a *CFP* mRNA with an alpha-tubulin 5'-UTR and the 3'-UTR of interest—in this case, the ~780nt 3'-UTR of *PGKC*. The puromycin resistance marker mRNA (*PAC*, puromycin acetyltransferase) is trans-spliced using the signal from the *ACT* gene. This splice signal also directs polyadenylation of the *CFP* reporter mRNA,

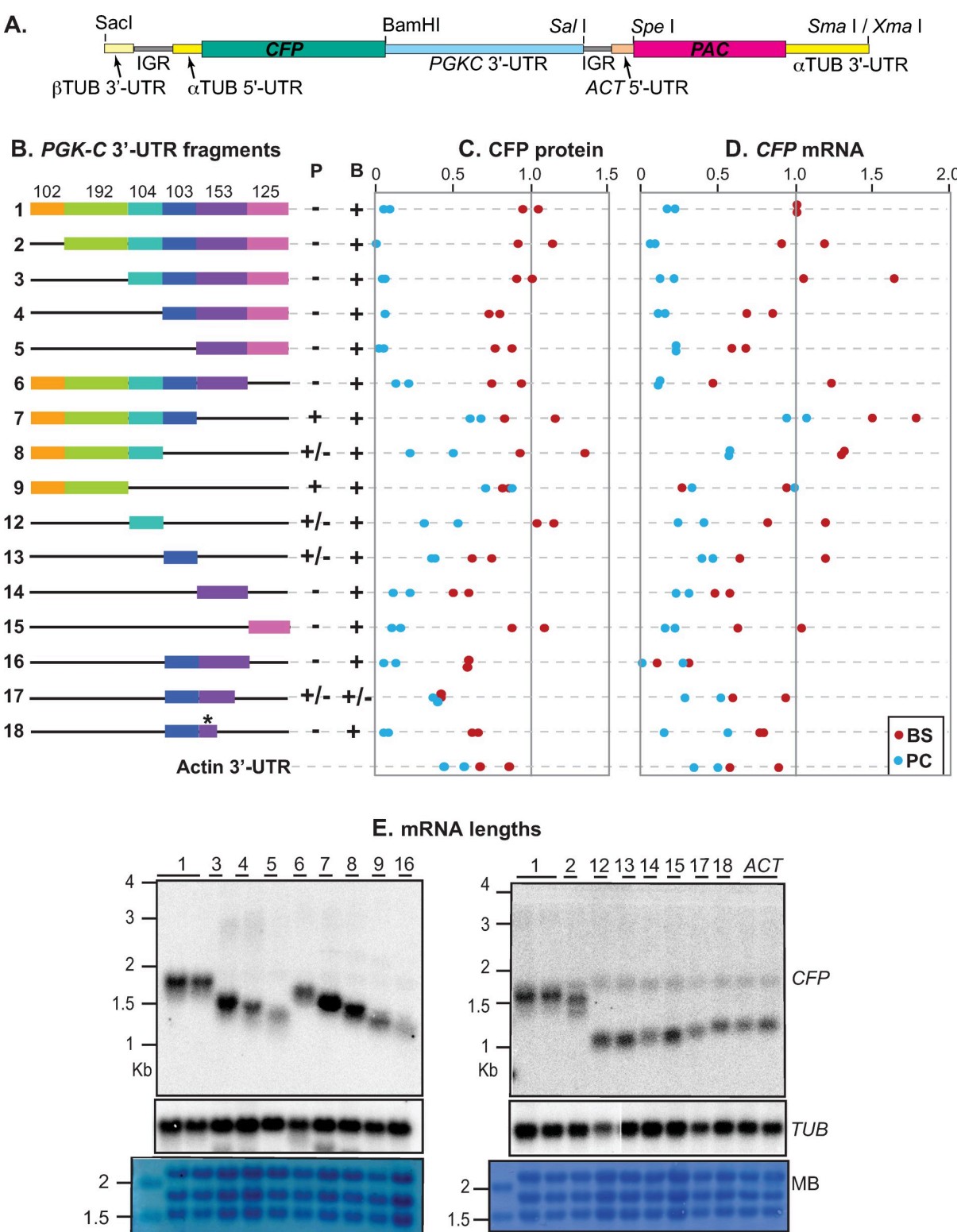

**Fig 3. The *PGKC* 3'-UTR contains at least two regulatory sequences. A.** Cartoon showing one of the *CFP* reporter constructs (pHD3261) used in this study. Different fragments of the *PGKC* 3'-UTR were cloned downstream of the *CFP* reporter coding sequence. The polyadenylation site "p (A)" is specified by the *PAC* splice signal. *TUB* denotes the tubulin locus. *ACT* denotes actin. **B.** Schematic diagram of the *PGKC* 3'-UTR segments used to map the regulatory sequences. Measurements are on the right. Each dot is a result for an independent clone, with red and blue representing bloodstream (B) and procyclic (P) forms, respectively. Protein levels were estimated from Western blot

quantitation and mRNA levels were measured by real-time PCR, taking the average of three technical replicates for at least two independent populations. Values were normalized to the arithmetic mean for the *PGKC* 3'-UTR control populations, bloodstream forms with pHD3261. For "expression", averages are: "-" = <0.25x, "+/-" = 0.25–0.5x, "+" = 0.5-2x, "++" = 2-3x and "+++" >3x. All available values for actin, including those from the experiment in Fig 4, were included in the current Figure. The Western blots for this and Fig 4 are at: https://figshare.com/articles/figure/Complete_Western_blots_for_PGKC_3_-UTR_assays/19188455 The qPCR results are at: https://figshare.com/articles/dataset/qPCR_for_PGKC_3_UTR_analysis/19188248 **C.** RNA from the bloodstream-form samples used in (B) were examined on Northern blots to examine the size of *CFP* mRNAs bearing *PGKC* 3'-UTRs. The methylene blue-stained membrane and the tubulin mRNAs are shown as loading controls. The gels ran slightly differently and the identity of the faint band at about 2 Kb in the right-hand blot is unknown, though it could be cross-hybridisation with the largest LSU rRNA. Measured mRNA lengths are in S2 Table. The original blots are at: https://figshare.com/articles/figure/Complete_Northern_blots_PGKC/19181888.

approximately at the position of the *Sal* I site that divides the tested 3'-UTRs from the intergenic region (Fig 3A). A full sequence of the plasmid is included as S1 File. To design deletions (Fig 3B) we examined the predicted secondary structure of the 3'-UTR using the RNA fold web server (http://rna.tbi.univie.ac.at/cgi-bin/RNAWebSuite/RNAfold.cgi). The resulting predictions are not included here because they were not subsequently verified in any way. However, our deletions were designed to avoid disruption of predicted secondary structures. The full sequence of the *PGKC* 3'-UTR is in S2 Text. For each deletion, several plasmids were sequenced and the one that best matched the expected sequence was selected for further use. In some cases a few nucleotides were deleted or exchanged: an alignment that shows all of the 3'-UTR sequences tested is in S1 Fig. (Plasmids 17 and 18 are not included but had no mutations relative to plasmid 16.)

To test the functions of the 3'-UTR segments, we cut each plasmid to remove the plasmid backbone, transfected the DNA into bloodstream-form trypanosomes, and selected independent populations for further analysis. The transfection should normally result in parasites with a single copy of the plasmid into the alpha-beta-tubulin tandem repeat, although insertion of two or more copies would be possible. To allow for this, two different populations for each plasmid were selected for protein preparation, RNA preparation, and differentiation into procyclic forms. Protein levels were assessed for each population by quantitative Western blotting, with ribosomal protein S9 as a loading control (Fig 3C). RNA levels were measured in triplicate technical replicates for each clone, by reverse transcription and real-time PCR (Fig 3D). All values were normalized to the average of expression in bloodstream forms for the 782nt full-length *PGKC* 3'-UTR. The lengths of the mRNAs, measured in bloodstream forms (Fig 3E) mostly appeared to be around 100nt longer than the expected lengths, which were calculated assuming a poly(A) tail length of 60nt (S2 Table). Either we under-estimated the poly (A) tail length, or the Northern measurements are consistently a little too long. Using *CAT* reporters, in bloodstream forms the *PGKC* 3'-UTR gave twice as much mRNA and CAT activity as the *ACT* 3'-UTR [55], and there was no developmental regulation of the *ACT* 3'-UTR reporter. Using CFP as the reporter, the difference between *PGKC* and *ACT* in bloodstream forms was less marked, and the *ACT* 3'-UTR gave about 30% less activity in procyclic forms than in bloodstream forms.

Our results revealed that all fragments that included the final 278nt showed developmental regulation of protein levels: the amount of protein in procyclic forms was too low to measure accurately (Fragments 1–5; Fig 3C). For these 3'-UTR fragments, RNA in procyclic forms was 15%±6% of the bloodstream-form level. The difference in mRNA level was less than that seen previously using *CAT*, and also less than that seen with the mRNAs that include the *PGK* coding region. It was also rather more variable, as we had previously observed for RBP10 (Fig 3D). Perhaps the mRNA level is more sensitive to cell density than the protein level; and the coding region may also contribute [65,66]. We here therefore focus mainly on regulation of protein expression. Bloodstream-form levels of expression for fragments 4 and 5 were slightly lower

than for the full-length plasmid (Fig 3C). Next, we deleted from the 3'-end (Fig 3B, fragments 6–9). Deletion of the final 278nt (Fig 3B, fragment 7) was sufficient to abolish developmental regulation (Fig 3C). Intriguingly, further deletion (Fragment 8) gave a plasmid which again showed a roughly 2-fold difference between bloodstream and procyclic forms, while an additional deletion (Fragment 9) resulted in no regulation.

The results so far localized the regulatory sequences to the last 278 nt of the 3'-UTR. Testing of individual fragments, however, gave a more nuanced picture. Fragment 12 gave regulation like fragment 8. Fragment 13 contains a UA (U)$_6$ motif, but fragment 13, if anything, gave slightly more expression in bloodstream forms than in procyclic forms. Fragments 14 and 15 independently suppressed expression in procyclic forms, indicating that they contain two separate regulatory elements, although fragment 15 was the most effective. Fragment 16—which includes fragments 13 and 14—behaved like fragment 14. We next made 3'- deletions of fragment 16. Intriguingly, an initial trucation (fragment 17) gave a 3'-UTR with no regulation at all, but further removal (plasmid 18) restored the regulation. Plasmid 18 contains just 61nt of fragment 14 (designated with an asterisk). The combined results suggest that the 61nt in plasmid 18 include a region that represses expression in procyclic forms, but whose function is affected by the surrounding sequence. Additional regulatory elements are in fragments 12 and 15.

## AU-Rich elements affect expression

We next looked for regulatory motifs that could be tested in another sequence context. First, we searched for sequences that might specifically repress expression in procyclic forms. We compared different sets of sequences using MEME or DREME [67,68]. We initially searched for 6–12 nt motifs enriched in the *RBP10* regulatory fragments relative to the those that lacked regulation. (The maximum was set because most RNA-binding proteins are specific for less than 12nt.) No motifs were found but the sample size was small, so we expanded the dataset by including 3'-UTRs from other co-regulated bloodstream-form specific mRNAs: the two regulatory regions for *PGKC*, Phosphofructokinase (Tb927.3.3270), pyruvate kinase (Tb927.10.14140), glycerophosphate isomerase (Tb927.1.3830, the hexose transporter THT1 (Tb927.10.8450) [69], and an aquaglyceroporin (Tb927.10.14160) - 11 sequences altogether. As a comparator we used 402 3'-UTRs from procyclic-specific mRNAs, of the same average length. No motifs with statistically significant enrichment were found in the bloodstream-form mRNAs.

Next, we looked at low-complexity regions which might serve to bind multiple copies of sequence-specific RNA-binding proteins. One element consists of (AU) repeats, present in fragments 1.2 ((AU)$_{11}$), 1.3 ((AU)$_9$ and (AU)$_8$), and 3.2.1 ((AU)$_{10}$), and of course the larger fragments that include them. We had some previous preliminary data that suggested that this sequence was implicated in good expression of ZC3H11, so we tested its function in the context of the *RBP10* 3'-UTR. Deletion of the AU repeat from *RBP10* fragment 1.2 indeed resulted in a drastic decrease of reporter expression levels in bloodstream forms (Fig 4A). Further, several fragments of the *RBP10* 3'-UTR contain poly (A) tracts (F1.2, 2.1, 2.2.1, 2.2.2, 2.2.3, 3.1.2 and 3.2.1), and we speculated that they might act by recruiting a poly(A) binding protein. However, deletion of the poly (A) tracts from one of these fragments unexpectedly resulted in an increase in reporter expression levels in bloodstream forms, rather than a decrease (Fig 4B).

A brief search of annotated 3'-UTRs revealed (AU)$_9$ or longer in 580 other mRNAs; this is almost certainly an under-estimate because many annotated 3'-UTRs are truncated. In order to find out whether either the repeat could enhance expression in another context, we inserted it between the *CFP* coding region and the *ACT* 3'-UTR. We here analysed just two independent populations. There was no effect on expression in either form (Fig 4C). Similarly, the

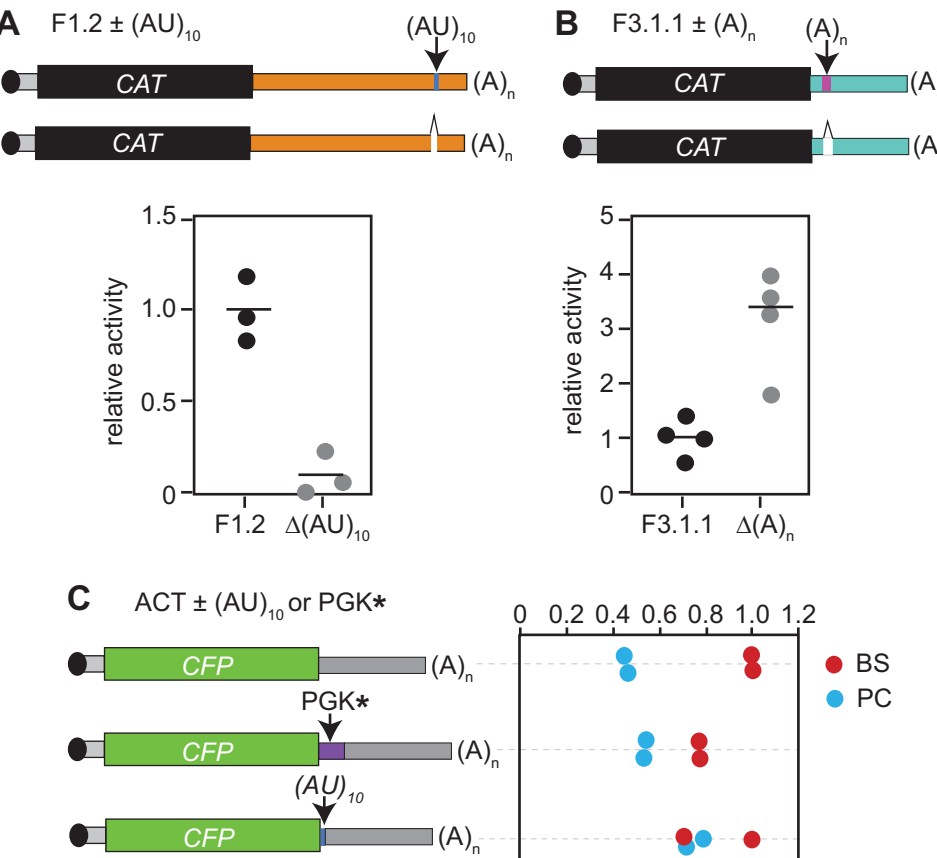

**Fig 4. Analysis of specific sequences. A.** Results for a *CAT* reporter plasmid containing *RBP10* 3'-UTR fragment 1.2 bearing $(AU)_{10}$ repeats and a mutated version without them. The average result for the fragment with the repeats is set to 1. **B.** As (A) but showing deletion of a poly (A) tract from fragment 3.1.1. **C.** Results for a CFP reporter plasmid containing the *ACT* 3'-UTR with either $(AT)_{10}$ or the PGKC* fragment inserted after the *CFP* coding region. The measurements are the averages of duplicate technical replicates for two independent populations, analysed with two different blots.

61nt fragment from the *PGKC* 3'-UTR (PGK*) was unable to repress expression in procyclic forms when inserted upstream of the *ACT* 3'-UTR. In both cases, therefore, the functions of these sequences depend on the surrounding sequence context.

## Discussion

The main aim of this study was to find sequences that can cause bloodstream-form specific expression in *T. brucei*, while a secondary aim was to investigate the general reason for the existence of very long 3'-UTRs. Regarding the first aim, we found that the *RBP10* 3'-UTR contains at least six sequences that specifically give low expression in procyclic forms (fragments 1.1, 1.2, 1.4, 2.2.2, 3 and 4) while the much shorter *PGKC* 3'-UTR contains at least two. There was also evidence, from the *RBP10* 3'-UTR, for sequences that stimulate expression in bloodstream forms (1.1, 1.2, 2.2.3, 3.1), or repress (3.2.2) or enhance (2.2.3) in both forms. We were unable to identify any specific short (<12nt) enriched motifs, either repeated within the *PGKC* and *RBP10* 3'-UTRs, or in comparison with other similarly regulated mRNAs. However, we were able to narrow down the regulatory regions considerably. The absence of a single enriched motif suggests that several different *trans*-acting factors, with different sequence

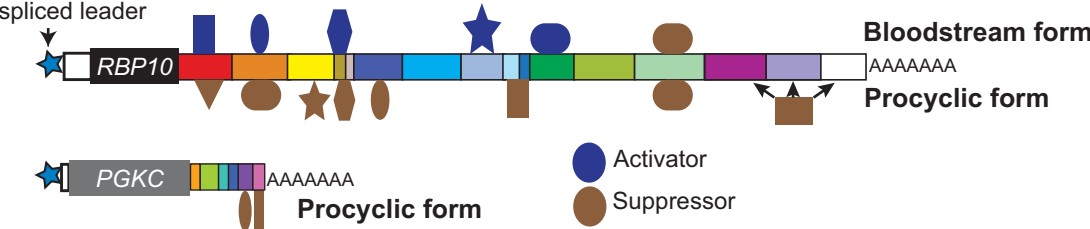

**Fig 5. Model for developmental regulation of *RBP10* and *PGKC*.** The two mRNAs are shown approximately to scale, using the colour coding from Figs 2 and 3. Regulatory RNA-binding proteins that bind the the 3'-UTRs are shown schematically with different shapes; it is possible that some of them bind in more than one position. Proteins that bind but do not give strong regulation, or which compete with other proteins for binding, are almost certainly also present but are not indicated.

specificities, are able to act independently to ensure appropriate expression of both PGKC and RBP10 (Fig 5). These factors probably each act on a subset of other similarly regulated mRNAs as well. Therefore, if our results are combined with further investigations of additional 3'-UTRs, it is likely that some short regulatory motifs will begin to emerge. Our results also suggested that the action of the regulatory motifs is context-dependent. For example, fragment 2.2.2 gave stronger regulation than the parent fragment 2.2, and fragment 3.1 gave higher activity in bloodstream forms than fragment 3 (Fig 2B). Perhaps the different fragments are bound by proteins that compete for binding on the *RBP10* 3'-UTR and this binding is influenced by secondary structures. Therefore, a dissection might allow some proteins to bind more efficiently while others are losing binding.

One interesting observation was the decrease in expression caused by deletion of (AU) repeats from one of the *RBP10* 3'-UTR fragments. This might mean that the repeats enhance expression, but if so, this is context dependent: when $(AU)_{10}$ was inserted at the 5'-end of the actin 3'-UTR it had no effect. Another possibility is that the (AU) repeats affect secondary structures of neighbouring regulatory elements. Our results nevertheless suggest that the functions of (AU) repeats should be considered during studies of other trypanosome mRNAs.

Many studies of mRNA binding proteins compare the mRNAs that are bound to that protein with the effects on the transcriptome after depletion of that protein. It is notable that in most cases, some bound mRNAs show clear changes in abundance after the protein is depleted, whereas others are unaffected. Our results clearly show why this is the case: for at least some mRNAs, regulation can be sustained by more than one region of the mRNA and probably, more than one RNA-binding protein. In practical terms, our results mean that screens for RNA-binding proteins that regulate one specific mRNA are unlikely to succeed if the reporter that contains the whole 3'-UTR is used for selection. For example, elimination of the regulatory protein that binds to fragment 15 of the *PGKC* 3'-UTR would probably have no effect on PGKC expression because regulation could be maintained by the sequences in fragment 14 (Fig 5). Indeed, a screen that was designed to find *trans* acting factors that regulate the mRNA encoding glycosyl phosphatidylinositol phospholipase C resulted only in the selection of 3' deletion mutants [70]. In order to search for proteins that regulate PGKC and RBP10 expression, it will probably be necessary to focus on the individual regulatory segments. Once such proteins are identified it is likely that they will be found to regulate other bloodstream-form-specific mRNAs as well.

The 3'-UTRs of the *RBP10* mRNA, and many other mRNAs encoding RNA-binding proteins, are extraordinarily long. Khong and Parker [56] have calculated that Opisthokont mRNAs are probably bound by at least 4–18 proteins/kb. If trypanosomes are similar, the *RBP10* 3'-UTR would be predicted to bind 28–126 proteins, while the *PGKC* 3'-UTR would bind between 3 and 14. Long mRNAs generally have relatively low abundances [15], which

might be important for regulatory proteins. The results presented here, however, suggest that long 3'-UTRs may have evolved as a "fail-safe" mechanism that ensures correct regulation even if the 3'-UTR is truncated by alternative processing, or one of the controlling proteins is absent.

## Supporting information

**S1 Table. Plasmids and oligonucleotides.**
(XLSX)

**S2 Table. mRNA lengths.**
(XLSX)

**S1 Fig. *PGKC* 3'-UTR sequence alignments.** The sequences were aligned using the Segman programme in the DNAStar package.
(PDF)

**S1 Text. *RBP10* 3'-UTR sequences.**
(DOCX)

**S2 Text. *PGKC* 3'-UTR sequence.**
(DOCX)

**S1 File. pHD3261 sequence in ApE format.**
(APE)

**S2 File. pHD3301 sequence in ApE format.**
(APE)

## Acknowledgments

We thank Dr. Monica Terrao for some initial analysis of the *RBP10* 3'-UTR. Six fragments of the *RBP10* 3'-UTR were cloned by Juyeop Kim, a master student, during his 6-week lab rotation. We thank Claudia Helbig and Ute Leibfried for technical assistance, for preparing media and buffers. We are indebted to Prof. Dr. Nina Papavasiliou (DKFZ, University of Heidelberg) and Prof. Dr. Luise Krauth-Siegel (BZH, University of Heidelberg) for allowing us to share their laboratories including equipment and reagents after the flood in ZMBH.

## Author Contributions

**Conceptualization:** Tania Bishola Tshitenge, Christine Clayton.

**Data curation:** Tania Bishola Tshitenge.

**Funding acquisition:** Christine Clayton.

**Investigation:** Tania Bishola Tshitenge, Lena Reichert, Christine Clayton.

**Methodology:** Tania Bishola Tshitenge, Bin Liu.

**Project administration:** Christine Clayton.

**Supervision:** Bin Liu, Christine Clayton.

**Visualization:** Tania Bishola Tshitenge, Lena Reichert, Christine Clayton.

**Writing – original draft:** Tania Bishola Tshitenge, Christine Clayton.

**Writing – review & editing:** Tania Bishola Tshitenge, Christine Clayton.

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
