## [Decision Letter · Decision Letter 0]

14 Feb 2022

Dear Prof. Clayton,

Thank you very much for submitting your manuscript "Several different sequences are implicated in bloodstream-form-specific gene expression in Trypanosoma brucei" for consideration at PLOS Neglected Tropical Diseases. As with all papers reviewed by the journal, your manuscript was reviewed by members of the editorial board and by several independent reviewers. The reviewers appreciated the attention to an important topic. Based on the reviews, we are likely to accept this manuscript for publication, providing that you modify the manuscript according to the review recommendations. 

I have evaluated the manuscript and the comments of the three expert reviewers. This work has clear merit and will contribute to our understanding of the mechanisms of gene expression regulation in trypanosomes, which is in line with the recommendations of the reviewers. In my view, the reviewers are clear and specific, and will hopefully present a resource towards providing a revised version of the paper for further evaluation. Only some minor changes are needed, with inclusion of supporting blot analysis data as supplemental information as requested by reviewer 2. Once you've had a chance to consider the reviewers' comments, please let me know if you have any questions.

Sincerely,

Guy Caljon

Associate Editor

Anthony Papenfuss

Deputy Editor

I have evaluated the manuscript and the comments of the three expert reviewers. This work has clear merit and will contribute to our understanding of the mechanisms of gene expression regulation in trypanosomes, which is in line with the recommendations of the reviewers. In my view, the reviewers are clear and specific, and will hopefully present a resource towards providing a revised version of the paper for further evaluation. Only some minor changes are needed, with inclusion of supporting blot analysis data as supplemental information as requested by reviewer 2. Once you've had a chance to consider the reviewers' comments, please let me know if you have any questions.

Reviewer's Responses to Questions

**Key Review Criteria Required for Acceptance?**

**Methods**

-Are the objectives of the study clearly articulated with a clear testable hypothesis stated?

-Is the study design appropriate to address the stated objectives?

-Is the population clearly described and appropriate for the hypothesis being tested?

-Is the sample size sufficient to ensure adequate power to address the hypothesis being tested?

-Were correct statistical analysis used to support conclusions?

-Are there concerns about ethical or regulatory requirements being met?

Reviewer #1: The methods are orthodox and well described, there no problems as far as I can tell.

Reviewer #2: The methodology is appropriate and the experiments are well designed.

Reviewer #3: No major issues with methods generally, but a few concerns about some of the stats used/not used

**Results**

-Does the analysis presented match the analysis plan?

-Are the results clearly and completely presented?

-Are the figures (Tables, Images) of sufficient quality for clarity?

Reviewer #1: Yes to all questions. The results are presented clearly, reflect the aims and support the conclusions.

Reviewer #2: The data are sound, however, a major concern is the lack of Western and Northern blots to illustrate the expression levels of the different cell lines in Figures 2, 3 and 4. These data should at least be included in the supplementary data.

Reviewer #3: Yes, analysis is logical, figures mostly of suitable quality/clarity

**Conclusions**

-Are the conclusions supported by the data presented?

-Are the limitations of analysis clearly described?

-Do the authors discuss how these data can be helpful to advance our understanding of the topic under study?

-Is public health relevance addressed?

Reviewer #1: The conclusions are appropriate and supported, as far as I can judge. There is no immediate public health relevance.

Reviewer #2: The conclusions drawn by the authors are consistent with the data presented.

Reviewer #3: conclusions are perhaps a little short and worthy of further exploration/explanation

**Editorial and Data Presentation Modifications?**

Reviewer #1: In places, I wasn't sure why various facts were being provided. I suspect the text could be clarified with a further redraft, with the authors asking themselves if they have explained everything they have written.

Reviewer #2: A few minor points that could also be addressed by the authors:

1. Line 217, “in order to find” instead of “in other find”

2. Line 251, remove one “that”

3. The authors used cyan fluorescent protein (eCFP) as reporter, but wrote “GFP protein” and “GFP mRNA” in figures 3C and 3D, respectively. Similarly, GFP instead of CFP is mentioned in S2 Table. This needs to be corrected. In addition, the authors must choose between CFP and eCFP throughout the manuscript.

4. In line 287, it is mentioned that “protein levels were assessed for each population by quantitative Western blotting, with ribosomal protein S9 as a loading control”, however the Western blots are not shown in the manuscript and should be included in the supplementary data (see general comment).

5. In the legend of Figure 3B, reverse blue and red in the sentence “Each dot is a result for an independent clone, with blue and red representing bloodstream (B) and procyclic (P) forms, respectively” to be consistent with the colors used in the figure.

6. In Figure 3A, all lanes showed an additional band of 1.7-1.8 kb detected by the CPF probe, especially in the right blot. In addition, the position of the markers is different between the two panels (see for example the mRNA detected in lanes 1 that is not have the same size in the left and right blots).

7. Line 325, “dataset” instead of “datset”.

8. In the second last paragraph of the result section, as well as in the corresponding figure 4, it is not mentioned in which form of parasite the experiments were conducted, PCF or BSF. We can assume it’s in BSF, but it should be clarified.

9. In the first column of figure 4B, which may correspond to the actin control in Figure 3B, one may expect to see an expression level in PCF around 0.5. But no value for PCF is included in this graph. This important control should be included in the figure.

Reviewer #3: One typo spotted I think, otherwise very clear.

**Summary and General Comments**

Reviewer #1: This study has delineated the 3'UTRs of two transcripts in Trypanosoma brucei that are developmentally regulated and encode important bloodstream-stage proteins. They have succeeded in producing a more accurate map of the regulatory motifs within the 3'UTRs, showing that they are likely bound by many different regulatory proteins. This is a useful achievement that progresses our understanding of the interaction hierarchies that must control development within these parasites. 

I have no direct experience of the methods used in the study, but they are certainly orthodox for the field and well described in the manuscript. I have no reservations about the quality of the work or the veracity of the findings. I have just three comments to make about the text.

First, it begins "Kinetoplastids are unicellular flagellated parasites that infect mammals and plants." I would suggest changing this since most Kinetoplastids are not parasites (better to use 'trypanosomatids') and trypanosomatids largely infect insects and vertebrates (although plants are included). "Mammals and plants" is too selective to be accurate.

Second, ll194-6 describe an acetyl-coA synthetase coding region downstream of the RBP10 gene. The authors state that "[it is] present elsewhere in the genome, but the presence of read alignments over the region that surrounds the Tb927.8.2790 coding region suggests that the pseudogene mRNA is also present in both bloodstream and procyclic forms." I didn't understand why we were being told this. Is the pseudogene implicated in the RBP10 3'UTR? Please explain what we need to know about this.

Third, in the section beginning on l319 the authors look for common sequence motifs among their regulatory sequences, but find none. Is a similar exercise not possible for secondary structures? Have the authors or others not looked for conserved loops or folds among these UTRs with similar expression profiles?

Reviewer #2: The 3'-UTR lengths in trypanosomes is remarkably long compared to other Opisthokont model organisms, with mRNAs having 3'-UTRs of 5 kb or more. This is particularly true for mRNAs encoding RNA-binding proteins, which have vital regulatory functions. As their expression has to be particularly tightly controlled, the authors hypothesized that these long 3'-UTR may contain several regulatory sequences acting independently. To address this question, Tania Bishola Tshitenge and colleagues used two report systems (CAT and eCFP) to identify the regulatory sequences within the 3’-UTR of two mRNA that are responsible for the specific expression in the bloodstream forms of T. brucei (BSF), i.e. the 7.2 kb 3’-UTR of RBP10, an RNA binding protein previously characterized by the same research team, and the terminal 424 nt of the PGKC 3'-UTR that the authors previously identified as responsible for specific expression in BSF. They found that PGKC has at least two such sequences, while RBP10 has at least four which are scattered throughout the 7.2 kb 3’-UTR. The authors concluded that several different sequence motifs - and therefore, probably, a similar number of RNA-binding proteins - are implicated in controlling bloodstream-form-specific mRNA stability and translation.

The data are sound and the conclusions drawn by the authors are consistent with the data presented. However, a major concern is the lack of Western and Northern blots to illustrate the expression levels of the different cell lines in Figures 2, 3 and 4. These data should at least be included in the supplementary data.

Reviewer #3: I went through this line by line, comments and questions given below with their line numbers.

48/50/51/52 PGK, systematic gene names the first time the gene is mentioned please.

70: quantitate "much more"

75: RBP6 systematic name?

88: Opisthokont, is this not a rather diverse group about which to make comparative comments?

94: systematic name for ZC3H22?

96-97: what is meant by "better expressed"? Not quite clear here, no numbers either.

98: give the 8mer sequence

103-105: maybe actually give the sequences here for the reader!?

129: are the RBP10 3' UTR sequences identical in 427 and 1125? Was there a reason not to use 1125 genomic DNA as a template for the PCRs?

135-136: very ambiguous about how much of what was actually sequenced to confirm identity: this is important

188-192: I found this a bit confusing: are there TWO regions of UTR like sequence in separate locations in the genome, and if so, how can you be certain which one is responsible for the noted effects? Are there not PacBio sequence data available for 1125, which might help clear up this quandry, given it is perhaps central to how robust your conclusions/inferences might be?

206-207: see 188-192?

Fig 1A, units for Y axis?

Fig 1A, procyclics, wild/crazy observation, but it looks as if majority of transcripts may not have the long UTR sequence for 2780, perhaps polyadenylated much closer to the stop codon? Do you have any oligo-dT based 3' end sequence/PCR data for checking this out? Or ratio of read depths for the CDS and 5UTR in B or P on basis that if it has the CDS it should have the UTR. Might fit with your comment on line 232. Might depend on the characteristics of the RNAseq libraries whether that is meaningful; the sequence reads used in Seattle were pretty short and were aligned locally using bowtie2; are there no more recent data with larger insert, paired end reads?

Fig 1D, lower panel: two? sets of blue dots, are they the three replicates?, no error bars

209-210: range, SD of measurements? Saying "probably" without stats is meaningless!

217: typo

Fig 2: no indication what the different colours indicate? I think they are regions/constructs, but we dont see them all together until Fig 5

Fig 2: should show the low-complexity repeat regions

Fig 2: For "expression", averages are: "-" = <0.25x, "+/-" = 0.25-0.5x, "+" = 0.5-2x, "++"=2-3x and "+++" >3x. I have "problems" with these categories, with values less than 1 being included in the "+" category, surely that would be either "-" or if redefined, "+/-" perhaps?

241-242: why did you not just use the means, would be much more appropriate? Currently a very curious choice of metric that artificially maximises the result divergence, particularly pronounced when n is such a small number...

243: "mostly not normal": by what measure did you reach this conclusion?

248: use nuclease digestion to make fragments?

261: what are the sequences of the regulatory elements proposed?

274: was this approach done for RBP10 at all?

278: was the RNAfold re-run on these deleted lines, to check that things were still as you expected/predicted?

293-294: this is possibly worthy of further explanation/exploration?

300-301: is the variability seen across any other mRNAs you measured (if you did)

303: which is...?

331-340: were RNAfolds done for all the various fragments to try and draw a link to the observed experimental results?

331-340: these are all longer than what the MEME search was done with, so you may be missing things

331-340: what other genes have similar AU repeats in their 3'UTR (or, for that matter, perhaps also the 5UTR?)

362-365: perhaps, but what are the sequences in the fragments? Are there any footprint data for these regions from other studies?

References: I have not checked them.

PLOS authors have the option to publish the peer review history of their article (what does this mean?). If published, this will include your full peer review and any attached files.

Reviewer #1: No

Reviewer #2: No

Reviewer #3: No

Figure Files:

Data Requirements:

Reproducibility:

References

---

## [Editor Report · Decision Letter 1]

3 Mar 2022

Dear Prof. Clayton,

We are pleased to inform you that your manuscript 'Several different sequences are implicated in bloodstream-form-specific gene expression in Trypanosoma brucei' has been provisionally accepted for publication in PLOS Neglected Tropical Diseases.

Best regards,

Guy Caljon

Associate Editor

Anthony Papenfuss

Deputy Editor

Thank you for addressing all comments of the reviewers and for providing the additional data in the figshare links.

Congratulations with your work.

Sincerely yours,

Guy Caljon

---

## [Editor Report · Acceptance letter]

17 Mar 2022

Dear Prof. Clayton,

We are delighted to inform you that your manuscript, "Several different sequences are implicated in bloodstream-form-specific gene expression in * Trypanosoma brucei*," has been formally accepted for publication in PLOS Neglected Tropical Diseases.

Best regards,

Shaden Kamhawi

co-Editor-in-Chief

Paul Brindley

co-Editor-in-Chief
